# Recombination in Coronaviruses, with a Focus on SARS-CoV-2

**DOI:** 10.3390/v14061239

**Published:** 2022-06-07

**Authors:** Daniele Focosi, Fabrizio Maggi

**Affiliations:** 1North-Western Tuscany Blood Bank, Pisa University Hospital, 56124 Pisa, Italy; 2Department of Medicine and Surgery, University of Insubria, 21100 Varese, Italy; fabrizio.maggi@uninsubria.it

**Keywords:** Coronaviridae, recombination, superinfection, coinfection, PANGOLIN, SARS-CoV-2, COVID-19

## Abstract

Recombination is a common evolutionary tool for RNA viruses, and coronaviruses are no exception. We review here the evidence for recombination in SARS-CoV-2 and reconcile nomenclature for recombinants, discuss their origin and fitness, and speculate how recombinants could make a difference in the future of the COVID-19 pandemics.

## 1. Recombination in Coronaviruses Other Than SARS-CoV-2

Recombination represents a major contributor to RNA virus evolution [1] together with re-assortment (which exclusively operates in RNA viruses with segmented genomes). Recombination can occur both in segmented [2,3] and non-segmented RNA viruses [4,5,6,7] and avoids an accumulation of irreversible deleterious mutations typical of asexual reproduction (so called “Muller’s ratchet” [8]). “Donor” and “acceptor” are conventional terms used to refer to the strain represented in a greater and lesser amount, respectively. Recombination within different sublineages of the same virus invariably requires co-circulation and co-infection of the same host.

Recombination can be difficult to detect whenever the sublineages have minimal differences and requires whole-genome sequencing (WGS). Unlabeled private mutations can help track the spread of the recombinant lineage more easily: they are defined as private mutations that are neither reversions nor labeled (i.e., they are not mutations to a genotype that is known to be common in a clade) [9]. Deletions are generally considered useful landmarks for recombination because they are unlikely to be reverted (except through recombination), but they can spontaneously occur across different sublineages (convergent evolution) independently from recombination (as seen, e.g., by the NSP6 SGF- reported in Alpha, Beta, and Gamma variants of concern of SARS-CoV-2 [10]).

We review here former evidence for recombination in betacoronaviruses and then focus on SARS-CoV-2 recombinants.

Coronaviridae undergo both homologous recombination (HR) and nonhomologous recombination (NHR). Only a minority of recombinants are likely detected in surveys since most of them are unlikely to be fitter than the currently dominant strain.

A high frequency of HR occurs across all three coronavirus groups, e.g., in murine hepatitis virus [11,12,13,14], transmissible gastroenteritis virus [15], bovine coronavirus [16], feline infectious peritonitis virus [17,18], and infectious bronchitis virus (IBV) [19,20,21,22]. RNA recombination is thought to be similar to poliovirus [23]: in this scheme, the viral RNA-dependent RNA polymerase (RdRp) detaches from one template with the nascent RNA strand intact and resumes elongation at the identical or similar position on another template. Recombination in MHV was reported at levels as high as 25% [24], a record for RNA viruses.

By virtue of an RdRp template switch likely occurring during synthesis of the (−)-strand templates for subgenomic mRNA (sgmRNA) synthesis, coronaviruses generate a 3′-coterminal nested set of sgmRNAs sharing a 65–90 nt long common leader. One form of NHR that occurs between genomic and sgmRNA has been hypothesized to result from the collapse of the transcription complex during (-)-strand discontinuous transcription [25]. Such a disruption would leave a partial copy of the leader sequence within the genome near the junction between two genes. Remnants of leader RNAs were found in the genomes of wild-type HCoV-OC43 [26], and the HCoV-HKU1 genome contains two very significant segments of embedded leader sequence (Woo et al., 2005) [27,28,29].

Among human coronaviruses, recombination was first reported in 2006 for both HCoV-HKU1 [30] and HCoV-NL63 [31,32], then in 2011 for HCoV-OC43 (genotype D since 2004) [33], in 2004 for SARS-CoV [34,35], and finally in 2014 for MERS-CoV [36,37] (leading to at least five strains with parts from both humans and camels [38]). Nevertheless, recombination rates across the genome of the human seasonal coronaviruses 229E, OC43, and NL63 vary with rates of adaptation [39]. To date, there is no well-documented example of recombination between extant coronaviruses of different groups.

## 2. Recombination in SARS-CoV-2

### 2.1. Recombinant Origin of SARS-CoV-2

Li et al. initially showed in March 2020 that SARS-CoV-2’s entire receptor-binding motif (RBM) was introduced through recombination with coronaviruses from pangolins, possibly a critical step in the evolution of SARS-CoV-2’s ability to infect humans [40]. This was later confirmed by Zhu et al. in December 2020 [41]. However, more recently, using sliding window bootstrap (SWB) to highlight the regions supporting phylogenetic relationships, SARS-CoV-2 was defined as a mosaic genome composed of regions sharing recent ancestry with three bat SCoV2rCs recently discovered in the Yunnan region of China (RmYN02, RpYN06, and RaTG13) or related to more ancient ancestors in bats from Yunnan and Southeast Asia [42], with no evidence of direct recombination with pangolin viruses.

### 2.2. Super-Infection or Co-Infection with Different SARS-CoV-2 Lineages

SARS-CoV-2 can be named according to different phylogenetic systems, which can often but not always be reconciled. The Global Initiative on Sharing All Influenza Data (GISAID) phylogeny classifies clades with progressive letters (https://www.gisaid.org/index.php?id=208, accessed on 29 April 2022). The Phylogenetic Assignment of Named Global Outbreak LINeages (PANGOLIN) nomenclature uses an alphabetical prefix and a numerical suffix to identify descendants (https://www.pango.network/the-pango-nomenclature-system/statement-of-nomenclature-rules/, accessed on 29 April 2022). NextStrain uses a year-letter system (https://docs.nextstrain.org/projects/ncov/en/latest/reference/naming_clades.html, accessed on 29 April 2022). Finally, the WHO uses progressive Greek letters to dynamically identify variants of interests (VOI) or concern (VOC) (https://www.who.int/activities/tracking-SARS-CoV-2-variants, accessed on 29 April 2022).

A few months after the initiation of the COVID-19 pandemic, co-infections were documented without any evidence of recombination. The first detailed case was described in February 2021 as co-infection from NextStrain 20A and 20B lineages, which was followed up for kinetics of relative abundance: a Portuguese patient had a prolonged viral shedding case (97 days long), first with a severe disease manifestation followed by a short second hospitalization episode, in an otherwise healthy young female [43]. More cases soon followed: e.g., co-infection by B.1.1.248 (either as major or minor haplotype) and B.1.1.33 or B.1.91, respectively [44], or co-infection between B.1.1.7 and B.1.351 [45] or GH and GR [46]. A less conclusive case of co-infection was reported from Iraq, suggesting the need for helper strains from defective co-infective strains [47]. A large study identified 53 (~0.18%) co-infection events (including with two Delta sublineages) out of 29,993 samples: apart from 52 co-infections with two SARS-CoV-2 lineages, one sample with co-infections of three SARS-CoV-2 lineages was firstly identified [48]. Another study identified coinfections in around 0.61% of all samples investigated (nine cases) [49].

Co-infections should be distinguished from subclonal variants (so-called intra-host evolution or quasi-species swarm), which naturally occur during infection, especially long-lasting infections in immunocompromised recipients either spontaneously or after selective pressure from antiviral therapeutics [50].

### 2.3. Evidence for Recombination in SARS-CoV-2

There is both *in silico* [51] and in vivo [52] evidence for recombination of different SARS-CoV-2 strains. Studies relying on linkage disequilibrium identified that SARS-CoV-2 recombination occurs at very low levels [52,53,54] or does not occur at all [55,56,57,58,59,60]. Several alternative methods are available for reconstructing genealogies explicitly in the presence of recombination, both with [61] and without [62,63,64] making the parsimony assumption, but none is tailored to the particular problem of detecting recombination in the presence of recurrent mutation. In fact, many tests of recombination assume that all mutations can only occur once at each site, and hence, recurrent mutation from convergent evolution (as it occurs in SARS-CoV-2) and systematic errors can confound signatures of recombination [7,27,36,65].

Hence, novel methodological approaches have been developed to detect recombinant genomes in SARS-CoV-2 lineages. Ignatieva et al. proposed a parsimony-based greedy heuristic algorithm for reconstructing plausible ancestral recombination graphs (KwARG) [66]: it does not scale well to large datasets but was powerful enough for disentangling the effects of recurrent mutation from recombination in the history of a sample [67]. Turakhia et al. developed Recombination Inference using Phylogenetic PLacEmentS (RIPPLES) to break the sequence into distinct segments that are differentiated by mutations on the recombinant sequence and separated by up to two breakpoints: for each set of breakpoints, RIPPLES places each of its corresponding segments using maximum parsimony to find the two parental nodes—hereafter termed donor and acceptor. RIPPLES is very fast with a large dataset but is biased against identifying recombination events near the edges of the viral genome. They identified 606 recombination events by investigating a 1.6M sample tree, showing that approximately 2.7% of sequenced SARS-CoV-2 genomes have recombinant ancestry, that recombination breakpoints occur disproportionately in the Spike protein region, and that cases were coinfected with 2–3 SARS-CoV-2 variants on average [68].

Haddad et al. observed recombination between different strains only in North American and European sequences [69].

Table 1 summarizes the recombinants between VOCs detected in more than one case (generally > 50 GISAID sequences). Many more cases are likely to have occurred between non-VOCs in a pre-VOC era or within individual hosts, such as a recombinant between B.1.160 and Alpha variants in a lymphoma patient chronically infected for 14 months [70]: nevertheless, those recombinant have been not fit enough to spread and outcompete the dominant strain of the moment.

Recombination has been proposed as a mechanism for the generation of B.1.1.7 (Alpha VOC) [71]. Accordingly, further recombination has been detected among B.1.1.7 and other strains (B.1.36.17, B.1.36.28, B.1.177, B.1.177.9, B.1.177.16, and B.1.221.1): interestingly, in six of eight instances (and all four of the transmitting groups, including one transmission cluster of 45 sequenced cases over 2 months), the mosaic viruses contain a Spike gene from the B.1.1.7 lineage, and in four instances, there is a proposed recombination breakpoint at or near the 5′ end of the spike gene [72].

As soon as the possibility of recombination emerged, nomenclature systems started considering how to name these sublineages. In the PANGOLIN phylogeny, all top-level lineages that are recombinants have a prefix that begins with “X” [73]. In most cases, they expect a minimum of 50 sequences to design a novel recombinant linage, but exceptions arise if the recombinant has a particular novelty or significance, with unusual breakpoint and/or parental lineages. As of 5 April 2022, CoV-lineages (https://cov-lineages.org/lineage_list.html) reports lineages from XA to XY, mostly from the UK (which contributes the vast majority of GISAID entries), suggesting new changes to nomenclature will soon be required. To date, neither WHO nor NextStrain phylogenies have a scheming name for SARS-CoV-2 recombinants.

We will here separately discuss recombination between SARS-CoV-2 VOCs.

### 2.4. Alpha-Delta Recombinants

Recombination between Alpha and Delta SARS-CoV-2 variants has, to date, been reported in a single case despite co-circulation from June 2021 to December 2021. Sekizuka et al. reported a Delta AY.29 and B.1.1.7 (later dubbed XC lineage) [75].

### 2.5. Beta-Delta Recombinants

Recombination between Beta and Delta has, to date, been reported in a single case despite co-circulation since December 2021. He et al. reported possible evidence of recombination in the Orf1ab (174–2692 and 5839) and Spike genes (21,801–22,281, previously proposed as a putative recombination region between the progenitor of SARS-CoV-2 and Bat-SL-CoV) in a patient (dubbed “49H”) maintaining a 1:9 Beta:Delta co-infection ratio for 14 days as part of an outbreak during a flight from South Africa to China [78].

### 2.6. Delta-BA.1 Recombinants

Delta and Omicron BA.1 co-circulated from November 2021 until February 2022: cases have been reported of Delta and Omicron co-infection [79,80]. Their recombinants are often colloquially referred to as “Deltamicron” or “Deltacron”. They were among the first recombinants to be named by PANGOLIN (XD, XF, and XS), but, as it happened for BA.1, all Deltamicron recombinant were soon out-competed by BA.2.

On 7 January 2022, virologist Leondios Kostrikis at the University of Cyprus in Nicosia deposited 52 sequences in GISAID, which were claimed by media as Deltamicron, but upon further inspection, these appeared to be due to laboratory artifacts (most likely laboratory contamination) or coinfections and were withdrawn from GISAID [81].

Ou et al. reported multiple additional amino acid mutations in the Delta Spike protein were also identified in the recently emerged Omicron isolates, which implied possible recombination events [82].

More individual cases of Deltamicron were reported, which do not have a PANGOLIN name designated yet, e.g.:Two clusters of apparent Delta-Omicron recombinants were identified in the United Kingdom (PANGO issue #422 and #441), which have a breakpoint upstream of Spike in orf1ab;Lacek et al. reported nine AY.119.2:BA.1.1 cases in the mid-Atlantic region of the USA, with breakpoints within the Spike gene (amino acids 158 to 339), containing substitutions common to Delta lineages at the 5′ end and Omicron lineages at the 3′ end [83];Leuking et al. reported two more cases in immunosuppressed patients (a 70-year-old male lung-transplant recipient and a 70-year-old female patient with uncontrolled diabetes) in Texas [84];Duerr et al. in New York reported an unvaccinated, immunosuppressed kidney-transplant recipient who had positive COVID-19 tests in December 2021 and February 2022 and was initially treated with sotrovimab. Viral sequencing in February 2022 revealed a 5’ Delta AY.45 portion and a 3’ Omicron BA.1 portion with a recombination breakpoint in the spike N-terminal domain, adjacent to the sotrovimab quaternary binding site [85];Bolze et al. identified two independent cases of infection by a Delta-Omicron recombinant virus in USA, where 100% of the viral RNA came from one clonal recombinant. In both cases, the 5′-end of the viral genome was from the Delta genome and the 3′-end from Omicron though the breakpoints were different [80].

Delta and BA.2 co-circulated minimally: accordingly, Delta-BA.2 recombinants only occurred in a doublet from the end of January in Sweden (PANGO issue #519) and a singlet again in January 2022 in Karnataka, India (PANGO issue #484). Another possible explanation for their scarcity is that countries with significant co-circulation (e.g., India and Philippines) do not perform WGS very frequently.

### 2.7. BA.1-BA.2 Recombinants

Most Omicron recombinants identified to date have the BA.1 as acceptor and the breakpoint within ORF1ab and hence preserve Spike protein from BA.2 (e.g., XE, XG, XH, XJ, XK, XM, XN, XP, XQ, and XR): this is not surprising since BA.2 currently outcompetes BA.1. XP is the lone exception, having BA.1.1 (the BA.1 sublineage with R346K mutation) as an acceptor (including Spike) and BA.2 as a donor. Among them, XE (also known as V-22APR-02 in Public Health England) is the most concerning, having a growth advantage over BA.2 estimated at first at +9.8% [86] and then raised to +20.9% (largely the same as observed for AY.4.2 over Delta in late 2021) [87]. This further increase in the basic reproductive number approaches SARS-CoV-2 as the most contagious virus in human history (see Figure 1).

Ou et al. identified, by scanning high-quality completed Omicron Spike gene sequences, 18 core mutations of BA.1 variants (frequency > 99%) (eight in NTD, five near the S1/S2 cleavage site, and five in S2). BA.2 variants share three additional amino acid deletions with the Alpha variants. BA.1 subvariants share nine common amino acid mutations (three more than BA.2) in the Spike protein with most VOCs, suggesting a possible recombination origin of Omicron from these VOCs. There are three more Alpha-related mutations (Δ69–70, Δ144) in BA.1 than in BA.2, and therefore, BA.1 may be phylogenetically closer to the Alpha variant. Revertant mutations are found in some dominant mutations (frequency > 95%) in the BA.1 subvariant [82].

Colson et al. in Marseille detected two samples with a recombinant genome that was mostly that of a BA.2 variant but with a 3′ tip originating from BA.1 [88]. Gu et al. in Japan reported two more cases with a breakpoint near the 5′ end of the Spike gene (nucleotide position 20,055-21,618) [89]. Leuking et al. in Texas reported two more cases in immunosuppressed patients [84].

## 3. Conclusions

Most recombinants to date have been reported in the UK, Denmark, and the USA mostly because those countries have more dense genomic surveillance programs. None of the recombinants detected so far seems to grow fast enough to become dominant, and greater concern comes from the emerging L452R-carrying BA.2 (e.g., BA.2.12.1 in New York or BA.2.11 in Bretagne) or BA.4/BA.5 sublineages. Albeit recombination is extremely likely to occur between SARS-CoV-2 lineages, several factors limit their generation and spread:(1)Pandemic waves from recent VOCs are becoming shorter and shorter, minimizing the time of co-circulation of different VOCs.(2)Apart from immunocompromised hosts, the duration of within-host viral replication is limited, again minimizing the room for co-infection/super-infection.(3)The increasingly high reproductive number achieved by the currently dominating VOC (BA.2) creates a major barrier for any novel strain to emerge (Figure 2). While approaching the asymptote of the reproductive number, only marginal gains in transmissivity will be possible. In this regard, many GISAID-powered bioinformatics tools are available (e.g., Cov-Spectrum [90] or SARS-CoV-2 Recombinant Finder [91]).(4)Detecting a recombinant lineage requires WGS efforts to stay in place given that, as for XE, Spike gene sequencing is not enough to detect recombination.

Nevertheless, even extremely rare events are likely to happen under massive viral circulation. In particular, we should not forget that COVID-19 is panzootic, and the possibility of recombination between an animal-adapted lineage and a human-adapted lineage could have unpredictable consequences on the efficacy of current COVID-19 vaccines.

## Figures and Tables

**Figure 1 viruses-14-01239-f001:**
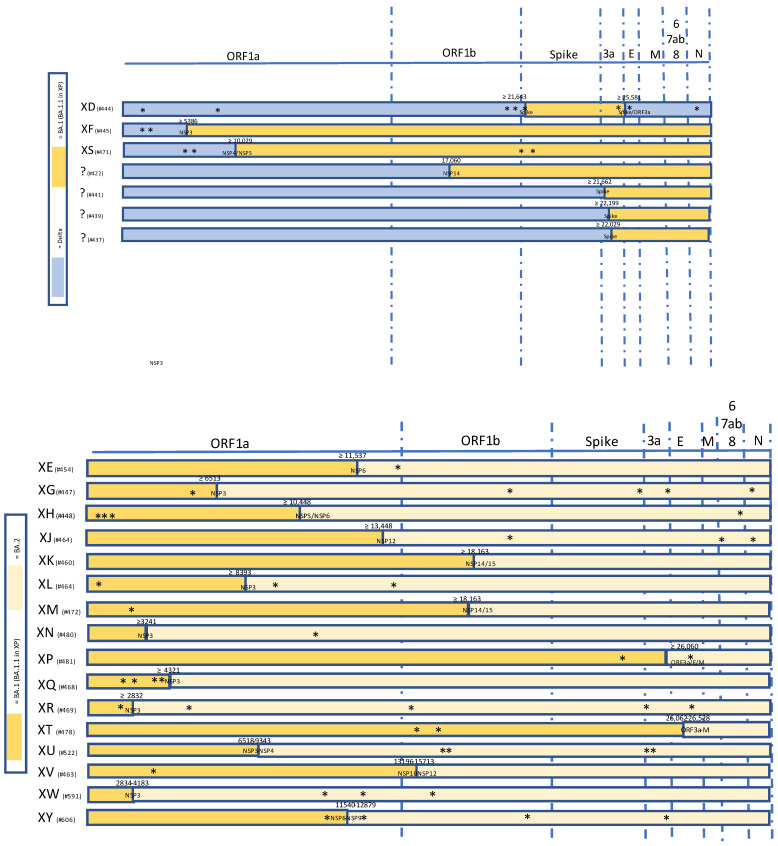
Summary of selected recombinant SARS-CoV-2 lineages of interest between VOC Delta and Omicron (upper panel) and within Omicron VOC sublineages (lower panel). Unlabeled private mutations are marked with *.

**Figure 2 viruses-14-01239-f002:**
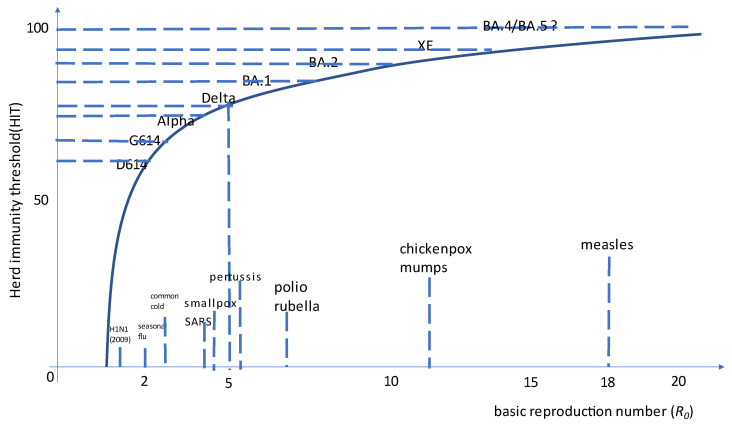
Basic reproductive number (*R*_0_) and estimated herd immunity threshold for SARS-CoV-2 variants of concern and the XE recombinant compared to other human pathogens. Please note herd immunity cannot be currently achieved with the current generation of systemically delivered vaccines [92].

**Table 1 viruses-14-01239-t001:** Details of PANGO-assigned recombinant SARS-CoV-2 lineages (modified from https://cov-lineages.org/lineage_list.html). More lineages can be found in Sakaguchi Hitochi’s Google Spreadsheet freely available online at https://docs.google.com/spreadsheets/d/1cQILRxXD756gJoRsaqMdJkxZm7sEjhV7ceY398Iz7gI/htmlview#gid=0 (accessed on 7 Jun 2022).

Name	Most Common Countries	Earliest Date	Parental Lineages		PANGO Designation Issue (Ref)	Unlabeled Private Mutations
			Donor	Acceptor		
XA	UK 51.0%, USA 42.0%, Czech Republic 2.0%, Sweden 1.0%, Switzerland 1.0%	18 December 2020	B.1.1.7 and B.1.177 (20E/EU.1)		n.a.	n.a.
XB	USA, Mexico, Guatemala, Honduras, El Salvador	8 July 2020	B.1.634 and B.1.631. Formally B.1.628		#189 [74]	n.a.
XC	Japan	12 August 2021	Delta AY.29 and B.1.1.7		#263 [75]	C27972T, G28048T, A28111G (ORF8: Q27, R52I, Y73C)
XD	France (40), Denmark (8), Belgium (1)	13 December 2021	Delta AY.4	Omicron BA.1 Spike (nt 21,643 to 25,581; codons 156–179)	#444 [76,77]	A1321C, A8723G, C20032T, G21641T, T21760C, C25667T, G25855T, G29540A (NSP2: E172D)
XE	UK (763) and Ireland, growth rate of +9.8% per week, with a growth advantage over BA.2 of ~ 10%	16 January 2022	Omicron BA.1	Omicron BA.2	#454	C14599T (NSP12)
XF	UK (39), no sample after 14 February 2022	7 January 2022	Delta AY.4 (or AY.4.x)	Omicron BA.1 (break point near the end of NSP3 at nt 5386)	#445	T1390C A2255G
XG	Denmark, UK, USA, Germany	11 January 2022	Omicron BA.1	Omicron BA.2 (likely breakpoint between 5927 and 6511 (NSP3))	#447	G5672T, A19855G, C25672T, G26062C, dG29140T
XH	Denmark	30 December 2021	Omicron BA.1	Omicron BA.2 (likely breakpoint between 10,448 and 11,287 (NSP5 or NSP6))	#448	T902C, C904A, G1244A, C28435T, (ORF1a:C213R, ORF1a:G327S, Nuc T902C)
XJ	Finland (Sweden, France, UK, Israel)	??-??-2022	Omicron BA.1	Omicron BA.2 (breakpoint seems to be between nt 13,200 and 17,400)	#449	T19857A, C27495T, -29759C
XK	Belgium (20)	13 February 2022	Omicron BA.1.1	Omicron BA.2	#460	(ORF1a:R1628C, ORF1b:Q866R)
XL	UK	6 February 2022	Omicron BA.1	Omicron BA.2	#464	G875T, T9208C, G14229A
XM	Europe	21 February 2022	Omicron BA.1.1	Omicron BA.2	#472	C2470T
XN	UK, Italy	1 February 2022	Omicron BA.1	Omicron BA.2 (likely breakpoint: between 2834 and 4183 at NSP3)	#480	G10986A
XP	UK	26 February 2022	Omicron BA.2	Omicron BA.1.1	#481	A24190C, C26880C
XQ	UK	13 February 2022	Omicron BA.1.1	Omicron BA.2	#468	A17615G, C2470T, (ORF3a:T223I, ORF1a:K856R, ORF1a:L3027F
XR	UK	13 February 2022	Omicron BA.1.1	Omicron BA.2 (likely breakpoint between 4322 and 4891 at NSP3)	#469	(ORF1a:K856R, ORF1a:T1543I, ORF1a:D4344N, A29510C)
XS	USA	19 January 2022	Delta AY.x	Omicron BA.1.1	#471	C5365T, C6196T, T13195C, C15240T, C21595T, C27807T
XT	South Africa (Gauteng, Limpopo and North-West)	13 December 2021	BA.1	BA.2 (likely breakpoint between 26,062 and 26,258 at ORF3a/M)	#478	C13994T, C16386T (S:G75V)
XU	India (Gurajat, Maharashtra), Japan, Australia	20 January 2022	BA.1	BA.2 (likely breakpoint between 6518 and 9343 at NSP3 or NSP4)	#522	C16887T, C17012T, C25416T, G25471C
XV	Denmark, Italy	31 January 2022	BA.1	BA.2 (likely breakpoint between 13,196 and 15,713 (NSP10 to NSP12))	#463	C3583T
XW	USA, Germany, England, Canada, Japan (ex-Finland)	13 March 2022	BA.1	BA.2 (likely breakpoint between 2834 and 4183 (NSP3)	#591	C10507T, C12756T, G16020T (ORF1a:T4164I)
XY	France, Israel, Scotland, USA	28 February 2022	BA.1	BA.2 (likely breakpoint between 11,540 and 12,879 (NSP6-NSP9)	#606	A1585G, T11049C (ORF1a:V3595A)

## Data Availability

All data are available at PubMed, medRxiv, and bioRxiv.

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
