# Peer review of "Recombination in Coronaviruses, with a Focus on SARS-CoV-2"

_viruses, 2022, doi:10.3390/v14061239_

Round 1
Reviewer 1 Report
Manuscript ID: viruses-1728846
Title: Recombination in coronaviruses, with a focus on SARS-Cov-2.
In this manuscript, the authors have reviewed the evidence for recombination in SARS-CoV-2.
Although the manuscript could be acceptable for publication in Viruses, several points must be clarified or corrected before resubmission.
- The Introduction should be improved.
The authors should develop the methods used to detect recombinant genomes and describe the mechanisms proposed to explain their occurrence.
- The paragraph “2. Recombination in SARS-CoV-2” should be renamed “Recombinant origin of SARS-CoV-2”.
The authors have focused on the putative pangolin origin of the RBM motif. However, the two references cited by the authors are a bit old (20: Li et al., 2020; 21: Zhu et al. 2020). As a consequence, the genomes recently discovered in bats (RmYN02, RpYN06, BANAL-20-52, etc.) were not included in the comparisons. Most recent studies have rather concluded that recombination occurred between bat viruses, with no evidence of direct recombination with pangolin viruses (https://www.mdpi.com/1999-4915/14/2/440). Therefore, I recommend to improve this paragraph by adding more recent studies showing that recombination occurred in bats, not in pangolins.
- In the paragraph “3. Super-infection or co-infection with different SARS-CoV-2 linages”, the authors wrote that “There is both in silico [22] and in vivo [23] evidence for recombination of different SARS-CoV-2 strains.”. This sentence is irrelevant in this paragraph. It should be moved to the beginning of paragraph “5. Evidence for recombination in SARS-CoV-2”.
- The paragraph “4. Nomenclature for SARS-CoV-2 recombinants” should be included at the end of paragraph 5.
- In paragraph 5, the authors should describe in details the new methodological approaches (Ignateva et al., 2021; Turakhia et al., 2021; etc.) used to detect recombinant genomes in SARS-CoV-2 lineages. The limitations of these methods should be also discussed.
- Since recombination between similar SARS-CoV-2 genomes can be very difficult to detect, the authors should provide a better description of the private mutations found in parental genomic fragments. I consider that these private mutations must be highlighted in Figure 1. For instance, I wondered how many private mutations were detected to show the double origin (delta or BA.1) of the three fragments of XD recombinants? and how many private mutations were detected to show the double origin (BA.1 or BA.2) of the two fragments of XE recombinants? etc. Possible mechanisms of recombination leading to either XE or XD recombinants could be also discussed.
- All recombinants described in paragraphs 6, 7 and 8 should be illustrated in a new figure similar to figure 1, showing private mutations found in parental genomic fragments.
- The Deltamicron “laboratory artifacts” mentioned in paragraph 8 need to be further explained.
Author Response
The authors should develop the methods used to detect recombinant genomes and describe the mechanisms proposed to explain their occurrence.
We have already explained in the introduction that whole-genome sequencing and occurrence of private mutations are required to study and identify recombinants. We have now much expanded the introduction to explain how homologous and nonhomologous recombination are in action in conoraviruses.
- The paragraph “2. Recombination in SARS-CoV-2” should be renamed “Recombinant origin of SARS-CoV-2”.
In the original submission, before MDPI reformatting, that was a level 1 title, with the following being level 2 titles. We have added a summary to clarify this. We have hence created the additional Level 2 title “Recombinant origin of SARS-CoV-2”under the Level 1 title “Recombination in SARS-CoV-2”.
The authors have focused on the putative pangolin origin of the RBM motif. However, the two references cited by the authors are a bit old (20: Li et al., 2020; 21: Zhu et al. 2020). As a consequence, the genomes recently discovered in bats (RmYN02, RpYN06, BANAL-20-52, etc.) were not included in the comparisons. Most recent studies have rather concluded that recombination occurred between bat viruses, with no evidence of direct recombination with pangolin viruses (https://www.mdpi.com/1999-4915/14/2/440). Therefore, I recommend to improve this paragraph by adding more recent studies showing that recombination occurred in bats, not in pangolins.
We have accordingly classified the pangolin evidences as early, and explained more recent evidences supporting bat origin.
- In the paragraph “3. Super-infection or co-infection with different SARS-CoV-2 linages”, the authors wrote that “There is both in silico [22] and in vivo [23] evidence for recombination of different SARS-CoV-2 strains.”. This sentence is irrelevant in this paragraph. It should be moved to the beginning of paragraph “5. Evidence for recombination in SARS-CoV-2”.
Moved as suggested.
- The paragraph “4. Nomenclature for SARS-CoV-2 recombinants” should be included at the end of paragraph 5.
Moved as suggested.
- In paragraph 5, the authors should describe in details the new methodological approaches (Ignateva et al., 2021; Turakhia et al., 2021; etc.) used to detect recombinant genomes in SARS-CoV-2 lineages. The limitations of these methods should be also discussed.
Added as suggested.
- Since recombination between similar SARS-CoV-2 genomes can be very difficult to detect, the authors should provide a better description of the private mutations found in parental genomic fragments. I consider that these private mutations must be highlighted in Figure 1. For instance, I wondered how many private mutations were detected to show the double origin (delta or BA.1) of the three fragments of XD recombinants? and how many private mutations were detected to show the double origin (BA.1 or BA.2) of the two fragments of XE recombinants? etc. Possible mechanisms of recombination leading to either XE or XD recombinants could be also discussed.- All recombinants described in paragraphs 6, 7 and 8 should be illustrated in a new figure similar to figure 1, showing private mutations found in parental genomic fragments.
We agree that private mutations are a key for identification and tracking of recombinants, but we feel it is not recommended to add extra figures, since the private mutations are already listed in Table 1 (column to the right). We are pointing reader to GitHub issues for detailed discussions of each recombinant.
- The Deltamicron “laboratory artifacts” mentioned in paragraph 8 need to be further explained.
We have specified laboratory contamination clarified as the most likely cause for these artifacts.
Reviewer 2 Report
This manuscript from Focosi and Maggi briefly reviews the current knowledge on recombination in Sars-CoV-2 infections. The manuscript is adequately written, but is shallow in its presentation of the subject. The title, "Recombination in coronaviruses, with a focus on SARS-CoV-2", is especially misleading, as the authors expend literally 9 lines (39-47) on coronaviruses other than SARS-CoV-2.
There are several major issues that need to be addressed to make this manuscript suitable for publication.
1. Recombination in other coronaviruses need to be described more extensively, including a presentation of the molecular mechanisms underlying recombination in these viruses. This will be relevant to explain the structure of the observed Sars-Cov-2 recombinants described in the following sections.
2. Section 2 is too short, and irrelevant as it is. It should be integrated into section 5 and introduced after describing the virus molecular epidemiology.
3. Sars-Cov-2 variants should be introduced in section 2, described properly (possibly with illustrations of specific mutations) and their nomenclature used consistently throughout the text (right now it goes back and forth between number coding and greek naming).
4. There seems to be confusion between circulating recombinants and singly identified recombinants in some parts of the text. For the most part the authors describe individually identified recombinants, that prove the recombination capability of this virus. However, in some instances (for example lines 134-5) it seems as specific recombinants were circulating in the population and were later replaced by a new strain. If that is the case it needs to be explained better, also clarifying the difference between circulating and non circulating recombinants.
5. Impact of recombination of viral fitness is at times implied (lines 100-4 and 134-5) but never explained. The authors have to include an introduction on the effect and consequences of viral recombination, possibly in section 1, including its effect on fitness.
6. Lines 73-7 are inconsequential for the rest of the manuscript, unless better contextualized and expanded. References need to be included regarding the mentioned effect of treatment on the generation of intrapatient variants.
7. The authors mention often that recombination occur mostly within Spike.yet the structures presented in figure 1 show widely spread recombination breakpoints. A better, or complete selection of recombinants would help support the data presented. Also, the authors could speculate as to why that would happen, based on the accumulated knowledge on viral recombination. A genomic structure descriptor should be included in figure 1.
8. The authors correctly relate recombination to co-infection in section 3, however the requirement for double infection to obtain inter-strain recombination should be made clearer, and the fact that Sars-CoV-2 variants co-circulated for relatively short times in different regions of the globe should be highlighted. In this respect, it would also be important to contextualize the identified recombinants referenced from different studies in relation to their coutry of origin and the co-circulaiton of variants in the region. The point being that recombinants cannot arise without co-circulation of variants.
9. The authors could choose to be more critical regarding the data presented, or at least comment of the findings they present here. Some of the studies referenced had limited evidence for recombination, and should be reported with caution.
10. Fitness and reproductive number are not necessary linked, as the authors declare in 10.3. The reproductive number curve (figure 2), its calculation, and meaning needs to be properly introduced before the discussion. Also, in figure 2 there is no scale for the y-axis.
Author Response
The title, "Recombination in coronaviruses, with a focus on SARS-CoV-2", is especially misleading, as the authors expend literally 9 lines (39-47) on coronaviruses other than SARS-CoV-2. 1. Recombination in other coronaviruses need to be described more extensively, including a presentation of the molecular mechanisms underlying recombination in these viruses. This will be relevant to explain the structure of the observed Sars-Cov-2 recombinants described in the following sections.
We have now much expanded the section on recombination in coronaviruses other than SARS-CoV-2, including molecular mechanisms.
- Section 2 is too short, and irrelevant as it is. It should be integrated into section 5 and introduced after describing the virus molecular epidemiology.
As per Reviewer #1 suggestion, we have now moved section into a novel section entitled “Recombinat origin of SARS-CoV-2”, which has been expanded.
- Sars-Cov-2 variants should be introduced in section 2, described properly (possibly with illustrations of specific mutations) and their nomenclature used consistently throughout the text (right now it goes back and forth between number coding and greek naming).
In order to facilitate the reader, we have initiated Section 2 with a summary of the main nomenclature systems for SARS-CoV-2 variants used throughout the text.
- There seems to be confusion between circulating recombinants and singly identified recombinants in some parts of the text. For the most part the authors describe individually identified recombinants, that prove the recombination capability of this virus. However, in some instances (for example lines 134-5) it seems as specific recombinants were circulating in the population and were later replaced by a new strain. If that is the case it needs to be explained better, also clarifying the difference between circulating and non circulating recombinants.
We have now specified the criteria for naming recombinants in Table 1 (i.e., at least 50 sequences for PANGOlin) and have separately discussed in the text recombinants reported from single patients.
- Impact of recombination of viral fitness is at times implied (lines 100-4 and 134-5) but never explained. The authors have to include an introduction on the effect and consequences of viral recombination, possibly in section 1, including its effect on fitness.
We have made this clearer at the position you suggested.
- Lines 73-7 are inconsequential for the rest of the manuscript, unless better contextualized and expanded. References need to be included regarding the mentioned effect of treatment on the generation of intrapatient variants.
We have added references describing intrahost variation after therapeutics, but we feel the positioning is right. It serves as a caveat to discriminate it from co-/super-infection.
- The authors mention often that recombination occur mostly within Spike.yet the structures presented in figure 1 show widely spread recombination breakpoints. A better, or complete selection of recombinants would help support the data presented. Also, the authors could speculate as to why that would happen, based on the accumulated knowledge on viral recombination. A genomic structure descriptor should be included in figure 1.
The only instance where we say that recombination occurs mostly in Spike is when reporting the finding by Turakhia et al and Jackson et al. We agree that most recombinants designated in PANGOlin and described in Table 1 actually have likely breakpoints outside Spike and at random locations throughout the genome, so we are unable to speculate on any preferential pattern. We have completed the representation of currently PANGOLIN-designated SARS-CoV-2 recombinants by adding in Table 1 the recently designated XV, XW, and XY.
- The authors correctly relate recombination to co-infection in section 3, however the requirement for double infection to obtain inter-strain recombination should be made clearer, and the fact that Sars-CoV-2 variants co-circulated for relatively short times in different regions of the globe should be highlighted. In this respect, it would also be important to contextualize the identified recombinants referenced from different studies in relation to their country of origin and the co-circulation of variants in the region. The point being that recombinants cannot arise without co-circulation of variants.
That point is already fully acknowledged in the text. In our opinion, the VOC share in a given country at the time the recombinant was described is not a fundamental data to report, given that the pandemic waves have been pretty synchronous worldwide and shares will largely overlap across countries and recombinants.
- The authors could choose to be more critical regarding the data presented, or at least comment of the findings they present here. Some of the studies referenced had limited evidence for recombination, and should be reported with caution.
We are fully confident that the recombinants presented in Table 1 have been fully investigated before being designated as such. We think caveats are implicit when instead pointing at recombinants detected in single patients, which are separately discussed in the main text.
- Fitness and reproductive number are not necessary linked, as the authors declare in 10.3. The reproductive number curve (figure 2), its calculation, and meaning needs to be properly introduced before the discussion. Also, in figure 2 there is no scale for the y-axis.
We have reformulated the sentence to avoid usage of the confounding word fitness. A scale has now been added to y-axis in Figure 2.
Round 2
Reviewer 1 Report
The authors have greatly improved the structure of the manuscript and its content. However, I still consider that figure 1 can be significantly improved by showing (with asterisks) the position of private mutations. By doing this, the authors will be able to correct some errors in table 1 (for example: C227972T at line 4 is obviously wrong). In table 1, I recommend to indicate all amino acid changes in parentheses just after (not before; see line 5) the private mutations involved.
Reviewer 2 Report
The authors addressed most of the concerns raised by this reviewer, and significantly expanded the text with relevant information.
Improved figures also add to clarity and readability of the text.
